# Coevolution Pigeon-Inspired Optimization with Cooperation-Competition Mechanism for Multi-UAV Cooperative Region Search

**Delin Luo** [1],*[iD]**, Jiang Shao** [1]**, Yang Xu** [1]**, Yancheng You** [1] **and Haibin Duan** [2,3]

[1] School of Aerospace Engineering, Xiamen University, Xiamen 361005, China; shaojiang@stu.xmu.edu.cn (J.S.); xuyang0108@xmu.edu.cn (Y.X.); yancheng.you@xmu.edu.cn (Y.Y.)

[2] School of Automation Science and Electrical Engineering, Beihang University, Beijing 100191, China; hbduan@buaa.edu.cn

[3] State Key Laboratory of Virtual Reality Technology and Systems, Beihang University, Beijing 100191, China

[*] Correspondence: luodelin1204@xmu.edu.cn



**Featured Application: Authors are encouraged to provide a concise description of the specific application or a potential application of the work. This section is not mandatory.**

**Abstract:** In this paper, a dynamic two-stage closed search (DTSCS) scheme for the unmanned aerial vehicle (UAV) cooperative region search is designed, which satisfies the range constraint (RC) and orientation constraint (OC). The closed trajectory is composed of two coupling stages, the search stage and the return stage. The position and orientation at the end of the search stage are the starting cell and orientation of the return stage. In the first stage, a coevolution pigeon-inspired optimization (CPIO) algorithm based on the cooperation-competition mechanism is proposed for multi-UAV cooperative search. In the return stage, inspired by region searching and trajectory tracking, a search tracking (ST) approach is presented to obtain the lowest-cost path under OC. The simulation results show that: (i) $N_p = 5$ is the best prediction time step. (ii) CPIO algorithm performs better than the compared intelligent algorithms in region searching. (iii) ST has high tracking performance than other algorithms. (iv) The DTSCS scheme enables every UAV to make the best use of its fuel to cover more region and return to the airport within the RC, and the average range utilization of UAVs is 97% under the 3OC.

**Keywords:** multi-UAV cooperative search; dynamic two-stage scheme; closed search trajectory; range constraint; orientation constraint; coevolution pigeon-inspired optimization; cooperation-competition mechanism; the lowest-cost path

## 1. Introduction

In the wide-area, complex and changeable environment, multiple unmanned aerial vehicles (UAVs) cooperative control is a critical research field [1–4] of unmanned system. In particular, multi-UAV cooperative search (MUCS) can be applied to prevent forest fires, patrol the border and probe potential safety hazards in cities and other aspects. Therefore, the study of MUCS strategy has practical significance. In [5], Sujit uses *k*-shortest path algorithm to search the target in an uncertain environment, but this algorithm cannot adapt to the situation that the edge weights of the search cell changes with the number of UAV passes. In [6], Bertuccelli models the uncertainty in the environment as the prior probabilities in the region and uses the Beta distribution to predict the minimum number of times required by UAV to search the target. This method, however, only considers a single UAV.

Riehl transforms MUCS into a finite sequence of updates to a dynamic graph via sampling region with a high probability of target existence in [7], but ignores the orientation constraint (OC) of UAV. In [8], Tian maps the search rewards of the target to fitness function and proposes a coevolution genetic algorithm to search the random target under the OC of UAV. In [9], Hu designs a multi-agent mapping fusion scheme based on distributed control, which converges the individual search probability of agent to the whole searching region. Nevertheless, this method does not take into account the impacts of environmental conditions.

Currently, almost all achievements on MUCS have mentioned that the range limits the application of UAV. To our knowledge, however, there is no study of the search strategies of MUCS under the range constraints (RC). The motivation of this paper is to present a new search strategy considering the range constraint (RC) and the OC. The search path of UAV under RC should be a closed trajectory. UAVs set out from the airports and need to return to their respective airports after completing the search task. The search algorithm based on an intelligent algorithm has been proved to be superior to the traditional optimization algorithm in [10–12]. Intelligent algorithms start the optimization with a series of possible solutions, and their performances depend primarily on the parameter initializations. Therefore, the constant parameters are not entirely consistent with the evolutionary spirit of the algorithm itself [13–15]. The results of optimization may be premature, divergent or locally optimal. Duan presents the pigeon-inspired optimization (PIO) algorithm based on a dynamic size of solution agents in [16]. Map and compass operator and landmark operator are used respectively by the distance of the pigeons from the target. Although there are only two switching operators, the optimization ability of PIO has been verified to be better than other heuristic algorithms in many applications [17–19].

The grid map can accurately describe the size, location, and shape of obstacles. Eight non-obstacle neighbor cells are reachable nodes for every cell. Its center is the allowed waypoint [20,21]. The lowest-cost path in grid model corresponding to finding a suitable sequence of cells to move the UAV from the original cell to a goal cell such that its total accumulated expenditure is minimized. The least-cost path solved by Dijkstra algorithm [22] or A* algorithm [23] does not take into account the OC of the UAV. It is assumed that the cells sequence of the lowest $path_a$ without the OC is $\mathcal{P}_a(L_a) = \{L_a(1), ..., L_a(h), ..., L_a(N_a)\}(h = 1, ..., N_a)$, in which $L_a(h)$ is the cell that $path_a$ contains. After adding the OC, if an element of $\mathcal{P}_a(L_a)$ is not in the feasible region, $path_a$ will become a illegal path. Therefore, A* algorithm and other algorithms cannot find the lowest-cost path under the OC.

**Our contributions: (i)** We apply the concept of the closed search to MUCS for the first time, and design a dynamic two-stage closed search (DTSCS) scheme to realize the closed path search of UAV. The first stage is the search stage, and the second stage is the return stage. The pose, position and orientation, at the end of the search stage is the starting pose of the return stage. **(ii)** Inspired by the cooperation-competition relationship between subgroups within a population in nature, a CPIO algorithm based on the cooperation-competition mechanism is used as the search algorithm for MUCS in the first stage. Every subgroup pigeons is abstracted as one UAV. The cooperation-competition relationship between pigeons reflects the cooperative relationship between UAVs. **(iii)** In the second stage, a search tracking approach (ST) is proposed. The cells containing in the lowest-cost path without the OC are modeled as the key regions in the region searching. The basic PIO algorithm is used to obtain the lowest-cost path from the starting point to the goal point which satisfies the OC. Maximizing search rewards is equivalent to minimizing tracking errors.

The rest of the paper is organized as follows. Section 2 introduces the concept of the closed path and the basics of region searching. In Section 3, A CPIO algorithm based on the cooperation-competition mechanism is proposed for MUCS. A ST approach is given to obtain the lowest-cost path with the OC in Section 4. Section 5 describes the DTSCS scheme. Numerical simulations and analysis are drawn in Section 6. Concluding remarks are given in the final section.

## 2. Problem Formulation

The searching region is usually divided into grid cells. The target existence probability (TEP) represents the probability of target existence in a cell. Environmental uncertainty (EU) indicates the degree to which UAVs do not know about the environment. These two variables are regarded as prior information. Every cell is modeled as the key or non-key region, and then the search probability graph of the whole searching region is formed. UAVs accomplish the search task via environment perception and information interaction. This section includes three parts: the environment model, UAV kinematic model and the analysis of UAV closed search.

### 2.1. Region Searching Model

This subsection describes the basic mathematical model of region searching. Firstly, rasterizing the searching region, and then the environment information is modeled as the prior probability information. The TEP in the whole search environment is updated by Bayesian rule. And the environmental uncertainty of every cell is updated according to search number of UAVs.

### 2.1.1. Environment Modeling

UAVs are assumed to fly on a fixed plane above the searching region. $M$ UAVs search $N_t$ targets. The searching region $R$ is uniformly divided into $L_x \cdot L_y$ cells:

$$R = \{C(m,n) \mid m = 1,2,...,L_x; n = 1,2,...,L_y\}, \tag{1}$$

where $C(m,n)$ is the coordinate of cell $(m,n)$. The side length of a cell is the unit length. Discretizing searching time of $UAV_i$ ($i = 1,2,...,M$), $UAV_i$ can search a cell in one time step [24,25]. The real-time position of $UAV_i$ can be described by the geometric center of the cell it searches.

### 2.1.2. Probability Map Update

It is known that the TEP $P_e^{m,n}(k) \in [0,1]$ and the EU $\chi_e^{m,n}(k) \in [0,1]$. If $0 \le P_e^{m,n}(k) \le 0.3$ and $0 \le \chi_e^{m,n}(k) \le 0.3$, the cell (m, n) is modeled as a known region. If $0.3 < P_e^{m,n}(k) \le 0.7$ and $0.3 < \chi_e^{m,n}(k) \le 0.7$, the cell (m, n) is modeled as a non-key region. And if $0.7 < P_e^{m,n}(k) \le 1$ and $0.7 < \chi_e^{m,n}(k) \le 1$, the cell (m, n) is modeled as a key region. Dangerous areas, such as hostile radar detection regions, peaks, etc., are designated as no-fly zones. Bayesian rule is used to update whether a cell exists a target [26]. The update equations for the probability that a target exists but is not detected and that the target does not exist but is reported are written as follows:

$$P_e^{m,n}(k+1) = \frac{P_e^{m,n}(k)P_c^{m,n}(k)}{(P_c^{m,n}(k) - P_f^{m,n}(k))P_e^{m,n}(k) + P_f^{m,n}(k)}, \tag{2}$$

$$P_e^{m,n}(k+1) = \frac{P_e^{m,n}(k)(1 - P_c^{m,n}(k))}{(P_f^{m,n}(k) - P_c^{m,n}(k))P_e^{m,n}(k) + 1 - P_f^{m,n}(k)}, \tag{3}$$

where $P_c^{m,n}(k) \in [0,1]$ is the detection probability of the airborne sensor to the target [27]. $P_f^{m,n}(k) \in [0,1]$ is the false alarm rate, which indicates the probability that there is no target in the cell but a target is reported. $\chi_e^{m,n}(k)$ decreases with the search number of UAVs, satisfying the following equation:

$$\chi_e^{m,n}(k+1) = \frac{1}{2^{N_f(m,n)}}\chi_e^{m,n}(k), \tag{4}$$

where $N_f(m,n) \in \mathbb{N}$ is the number of times that a cell (m, n) searched by UAVs.

## 2.2. UAV Kinematic Model

The kinematic model of $UAV_i$ with constant velocity and OC is obtained as follows:

$$
\begin{cases}
\dot{x}_i(t) = v_i(t)\cos\theta_i(t), \\
\dot{y}_i(t) = v_i(t)\sin\theta_i(t), \\
\dot{y}_i(t) = 0, \\
\dot{\theta}_i(t) \leq \varepsilon_i, \\
\dot{v}_i(t) = 0,
\end{cases}
\tag{5}
$$

where $(x_i(t), y_i(t), z_i(t))$ is the position of $UAV_i$, $v_i$ is its velocity. $\theta_i$ is course angle. The fourth term sets the constraint on the angle rate, which cannot exceed $\varepsilon_i$. In the grid map, $UAV_i$ can only move to one of its eight neighbor cells at a time step, corresponding to the heading $D_i(k) = \{0, 1, 2, 3, 4, 5, 6, 7\}$ [28].

**Definition 1. (D-orientation constraint, DOC)**. *In the grid map, The degree of freedom of the heading in which the UAV is allowed to walk in the next time step is D.*

A slow-moving robot usually satisfies 8OC (equivalent to no constraint ) or 4OC , as shown in Figure 1a,b. The fast UAV can not turn backward or turn vertically during flying. Therefore, it is subject to 3OC: go straight (0°), turn left (45°), and turn right (−45°), which can be seen in Figure 1c. The orientation of the next time step of $UAV_i$ can be summarized as follows:

$$
D_i(k+1) = \{(D_i(k) - 1) \bmod 8, D_i(k) \bmod 8, (D_i(k) + 1) \bmod 8\}.
\tag{6}
$$

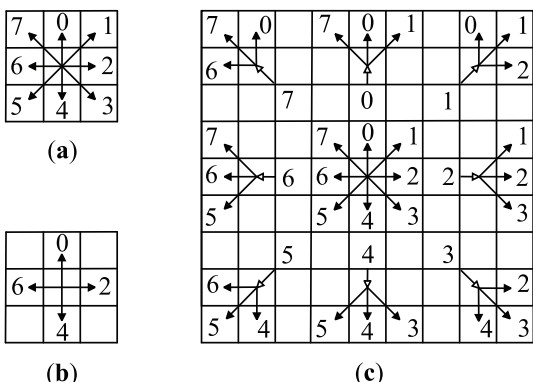

**Figure 1.** OC of motion: (**a**) 8OC; (**b**) 4OC; (**c**) 3OC.

## 2.3. Preliminary Analysis

In the research of MUCS, we always discuss the cooperative search strategy under the assumption that there is enough fuel, which is not consistent with the actual situation. Every UAV sets out from the airport and returns to its airport after completing the search task. The flight path is a closed track. We divide the closed search into two stages: search stage and return stage.

### 2.3.1. Closed Path Search

**Definition 2. (Closed path)**. *According to the definition of the loop in the directed graph, a closed path is a flight trajectory of a UAV starting form an airport to carry out task and returning back to the same airport.*

The closed search in this paper is different from the traveling salesman problem, which to find a solution that traverses all the reachable nodes from the starting point and returns to the starting point with the least travel cost. In this paper, the closed search refers to the closed path that UAV returns to the airport after completing the search task in the grid map.

Rasterizing searing region is consistent with discretizing searching time, so the RC of UAV is equated with its searching time limit. The closed path of the UAV under the RC should be composed of two stages: (1) search stage; (2) return stage. Long searching time means that UAV has a limited time to return and may not even return on time. And the short searching time means that the UAV can not effectively complete the search task. Ideally, the UAV would return to the airport at the time step when it is running out of fuel.

### 2.3.2. Search Stage

UAVs accomplish the search task via environment perception and information interaction. The principles of MUCS should include the following:

- Get the maximum TEP
- Acquire the maximum reduction of EU
- Avoid UAVs repeatedly searching the same cell
- Prohibit collisions between UAVs
- Fly as far away from the airport as possible to search more unknown cells
- Avoid searching for locally highly remunerated sequences of cells

### 2.3.3. Return Stage

**Definition 3. (The lowest path with *D*-orientation constraint)**. *The solution that the UAV seeks from origin to the goal cell with the smallest travel cost under the OC.*

The search strategy is no longer considered in the return stage. Our goal is to find a safe and collision-free lowest path under the OC. This path starts with the current cell and orientation and ends at the airport.

For the closed search, the primary purpose of this paper is to solve the following three problems: (1) Set the time step for UAV to return to the airport; (2) Maximize search rewards in the search stage; (3) The purpose of the return phase is to obtain a shortest path that satisfies the OC.

## 3. CPIO Algorithm with Cooperation-Competition Mechanism

According to the principle of cooperative search discussed above, we design the reward function of MUCS and regard it as the fitness function of PIO. In [8,29,30] coevolution genetic algorithm (GA) is applied to multi-robot platform to find the optimal solution through the cooperation among robots. In some conditions, the optimal solutions of every robots may sometimes be mutually exclusive. This conflict is consistent with the competition of subgroups in nature. Inspired by the cooperation-competition relationship between subgroups within a biological community in nature, a CPIO algorithm is used as the search heuristics for MUCS in the search stage. The optimal solution obtained via the CPIO algorithm is the position of a target, which is consistent with the attribute of region searching.

### 3.1. Design Reward Function

The MUCS process is modeled as a multiobjective nonlinear programming function with RC, OC, no-fly zone and collision free and the model is given in Equations (7) and (8):

$$J_1 = \omega_1 \sum_{q=1}^{N_p} P_e^{m,n}(k+q) + \omega_2 \sum_{q=1}^{N_p} \Delta\chi_e^{m,n}(k+q) + \omega_3 \sum_{q=1}^{N_p} F_r(k+q) + \omega_4 \sum_{q=1}^{N_p} F_a(k+q), \qquad (7)$$

$$s.t. \begin{cases} R_c \in R, \\ D_i(k+1) = \{(D_i(k) - 1) \bmod 8, \ D_i(k) \bmod 8, \ (D_i(k) + 1) \bmod 8\}, \\ T_{i,1}(k+1) < T_i, \\ d_{ij}(k), ..., d_{ij}(k + N_p) > 0, \end{cases} \tag{8}$$

where $\omega_1, \omega_2, \omega_3, \omega_4$ are weighting factors, which satisfy $\omega_1, \omega_2, \omega_3, \omega_4 \in (0, 1)$ and $\omega_1 + \omega_2 + \omega_3 + \omega_4 = 1$. $R_c$ is the searchable region in $R$. $T_{i,1}(k)$ and $T_i$ are the searching time and the range of $UAV_i$ respectively. $d_{ij}(k) = \sqrt{(x_i(k) - x_j(k))^2} + \sqrt{(y_i(k) - y_j(k))^2}$ denotes the Euclidean distance of $UAV_i$ and $UAV_j$. $N_p \geq 1$ is a positive integer. The first and second terms of the $J_1$ are designed to maximize search rewards, which corresponding to the first and second terms of the principles. The reduction of EU of the cell $(m, n)$ can be defined as follows:

$$\Delta \chi_e^{m,n}(k) = \chi_e^{m,n}(k+1) - \chi_e^{m,n}(k). \tag{9}$$

The third of the $J_1$ is consistent with the fourth of the principles and is designed to avoid collision between UAVs:

$$F_r(k) = \begin{cases} -100, & d_{\min}(k) \leq \sqrt{2}, \\ -\dfrac{1}{e^{d_{\min}(k)}}, & d_{\min}(k) > \sqrt{2}, \end{cases} \tag{10}$$

where $d_{\min}(k)$ is the minimum element of the $d_{ij}(k)$ at time step $k$, and $d_{ij}(k)$ is given in Appendix A Equation (A1). When the distance between any two UAVs is less than or equal to $\sqrt{2}$ unit lengths, the reward of this term will be $-100$. And then the rewards of $J_1$ will also be negative. $UAV_i$ can only choose to search cells away from other UAVs. With regard to the fourth term of the $J_1$, it implies that the further away $UAV_i$ is from the airport, the more rewards it obtains (principle 5):

$$F_a(k) = e^{-\dfrac{1}{d_\Sigma(k)}}, \tag{11}$$

where $d_\Sigma(k)$ is the sum of all the elements in $d_{ih}(k)$, which is written in Appendix A Equation (A2).

**Definition 4.** (*$N_p$-step-ahead prediction, $N_p$SAP*). *Learning from the idea of rolling optimization in model predictive control theory [31]. In each term of $J_1$ designed in Equation (7), the rewards of the reachable cells to be calculated is not only the next time step, but also the next $N_p$ time steps.*

**Remark 1.** *Although n steps are predicted in advance each time, $UAV_i$ flies forward only one-time step.*

The series of reachable waypoints of $N_p$SAP of $UAV_i$ maps the predicted cells sequence $\mathcal{E}_i(k) = \{\mathcal{L}_i(k+1), ..., \mathcal{L}_i(k+q), ..., \mathcal{L}_i(k+N_p)\}(q = 1, ..., N_p)$ from the step $(k+1)$ to the future time step $(k+N_p)$, which is based on the present position $L_i(k)$ at time $k$. The reachable cells included in 3SAP construct an expanding tree, as shown in Figure 2a. Obviously, the expanding tree generated by $N_p$SAP contains $3^{N_p}$ alternative paths and the $l$-th path can be illustrated as:

$$\mathcal{P}_i^l(k) = \{L_i^l(k+1), ..., L_i^l(k+q), ..., L_i^l(k+N_p)\}, \tag{12}$$

where $L_i^l(k+q) \in \mathcal{L}_i(k+q)$.

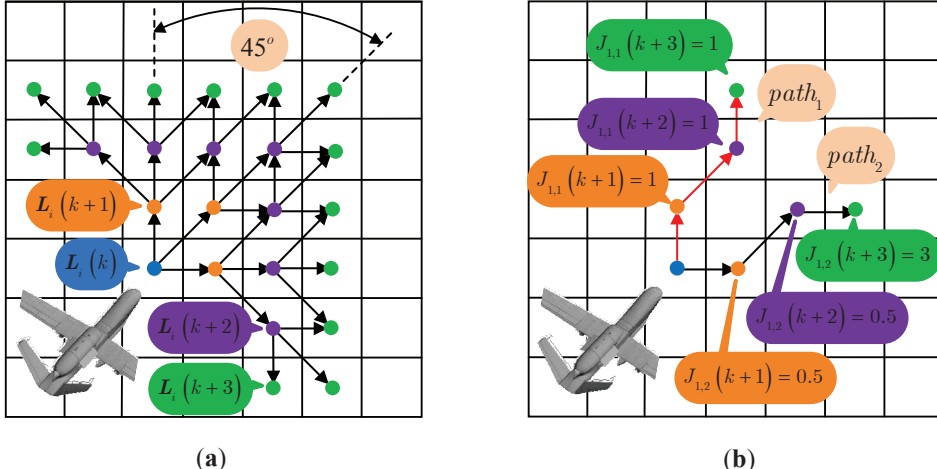

|       |       |
| :---: | :---: |
| (**a**) | (**b**) |

**Figure 2.** Schematic diagram of $N_p$SAP: (**a**) $N_p = 3$; (**b**) $N_p$SAP to avoid greedy thought.

The positive aspects of $N_p$SAP include:

- Avoid greedy thought corresponding to the sixth term of principles referred in Section 2.3.2
- Circumvent the no-fly zone ahead of time and continue to search the key region
- Prohibit collision between UAVs
- Reduce the impact of communication delays

As given in Figure 2b, Comparing the reward of $path_1$ and $path_2$, if $N_p = 1, J_{1,1}(k+1) > J_{1,2}(k+1)$, the $path_1$ is chosen at this time step. Else if $N_p = 3, J_{1,1}(k+1) + J_{1,1}(k+2) + J_{1,1}(k+3) > J_{1,2}(k+1) + J_{1,2}(k+2) + J_{1,2}(k+3)$, the $path_2$ will be chosen. For an extreme situation, if $N_p = T_{i,1}$, the path with the largest reward in $\mathcal{E}_i(k)$ must be the optimal path in the whole search process. As $N_p$ grows, the number of paths increases exponentially. Therefore, we use PIO algorithm to find the optimal solution that is difficult for conventional optimization algorithms.

### 3.2. Overview of Basic PIO

Inspired by natural phenomenon of the autonomous homing behavior of pigeon swarms, Duan proposes the PIO algorithm in [14], which the optimization process can be divided into two operators based on the distance of the pigeons from the destination.

**Operator 1: Map and compass operator**

During the prometaphase of the search, pigeons is far away from the goal. The real-time information of the sun is abstracted into map and compass operator to adjust the flight orientation, which is a rough navigation process. Suppose the number of pigeons is $C_1$. The position and velocity of $pigeon_a, (a = 1, 2, ..., C_1)$ in the two-dimensional (2D) plane is expressed as:

$$\begin{cases} L_a = [x_a, y_a], \\ V_a = [v_{x,a}, v_{y,a}]. \end{cases} \tag{13}$$

The renewal equations of position and velocity are:

$$\begin{cases} L_a^u = L_a^{u-1} + V_a^u, \\ V_a^u = V_a^{u-1} e^{-R_p \cdot u} + rand(L_{best} - L_a^{u-1}), \end{cases} \tag{14}$$

where $u = 1, 2, ..., N_{c1}$ is the current iteration number. $R_p$ is the coefficient of the map and compass operator. $rand$ is a random number from 0 to 1. $L_{best}$ denotes the position of the pigeon closest to the goal in the pigeons in iteration $u - 1$. After $N_{c1}$ iterations, the rough navigation stage is completed. PIO algorithm enters the second stage of optimization.

**Operator 2: Landmark operator**

When the pigeons arrive near the target, the PIO algorithm switches to the landmark operator. The landmark information of the nearby environment will provide precise guidance information. The speed of the pigeon does not change at this stage, while the position is updated:

$$L_{center}^{v-1} = \frac{\sum\limits_{a=1}^{C_2^{v-1}} L_k^{v-1} \cdot fitness(L_a^{v-1})}{C_2^{v-1} \cdot \sum\limits_{a=1}^{C_2^{v-1}} fitness(L_a^{v-1})}, \tag{15}$$

$$L_a^v = L_a^{v-1} + rand \cdot (L_{center}^{v-1} - L_a^{v-1}), \tag{16}$$

where $v = 1, 2, ..., N_{c2}$ is the current iteration number. $C_2^v = \dfrac{C_2^{v-1}}{2}$ is the number of pigeons in $v$ iteration. $L_{center}^{v-1}$ denotes the position of the central pigeon in $v-1$ iteration. $fitness(\cdot)$ is the fitness function. The optimal solution will be obtained after Operator 2 is performed $N_{c2}$ iterations.

*3.3. CPIO and Cooperative Search*

In biology, a population may divide into some subgroups [32,33]. In the face of natural enemies, the subgroups will cooperate to resist. Nevertheless, they also compete with each other in the interests of food, mating, territory, and so on. Cooperation and competition make the population survive and evolve better. To simulate these natural behaviors, UAVs work together to complete the search task via information interaction, in the mean time UAVs compete with each other to search the specific vital cells. We propose a CPIO algorithm based on the cooperation-competition mechanism as the search algorithm for MUCS, as shown in Figure 3. In which, one subgroup pigeons is abstracted as one UAV. The cooperation-competition relationship between pigeons reflects the cooperative relationship between UAVs.

MUCS based on CPIO is composed of three parts: CPIO, UAVs, and environment model. The principle of which is shown in Figure 4. In the CPIO module, the initial information of the environment, the environment information detected by the UAVs and the real-time pose signals of the UAVs are modeled as prior information. This information is then transmitted to the reward function $J_1$. $Subgroup_i$ transforms the optimal solution obtained by the cooperative-competitive mechanism into discrete flight signals. And these signals will be transmitted to $UAV_i$ at each time step. Thus, the cooperation-competition mechanism can be described as follows:

**Cooperation mechanism**

- Get the maximum $P_e^{m,n}(k)$ and the maximum $\Delta\chi_e^{m,n}(k)$
- Stay away from airports and search for more unknown regions
- Avoid searching locally highly remunerated cells

**Competition mechanism**

- Forbid UAVs to search for the same cell at the same time step
- Avoid UAVs repeatedly searching the same cell
- Stay away from other UAVs to search more unknown cells

Under all constraints, the subgroup that maximizes the total rewards will win. Cooperation and competition are not entirely opposed to each other. The winning subgroup will search the controversial cell, and the other subgroups compete to search other cells. The result of the competition mechanism is consistent with the original intention of the cooperation mechanism.

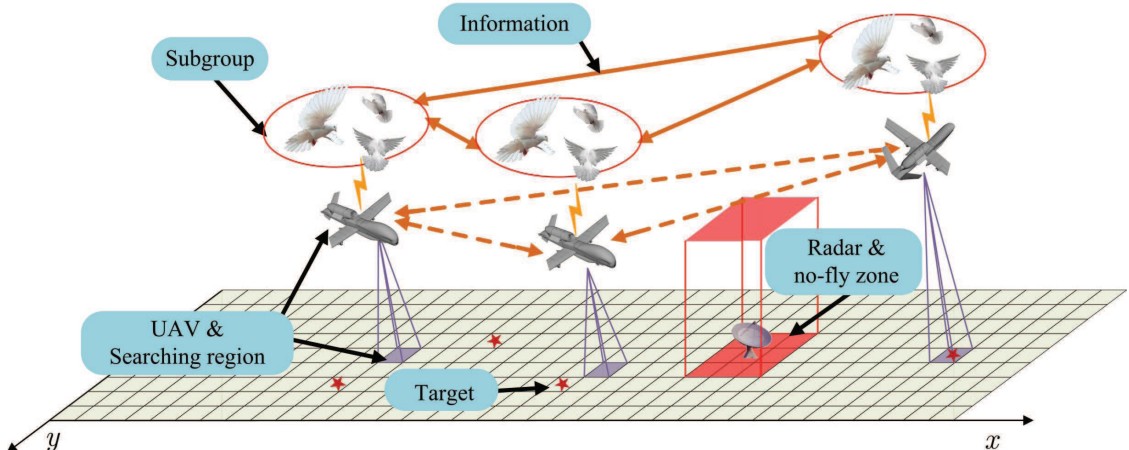

**Figure 3.** CPIO algorithm for MUCS.

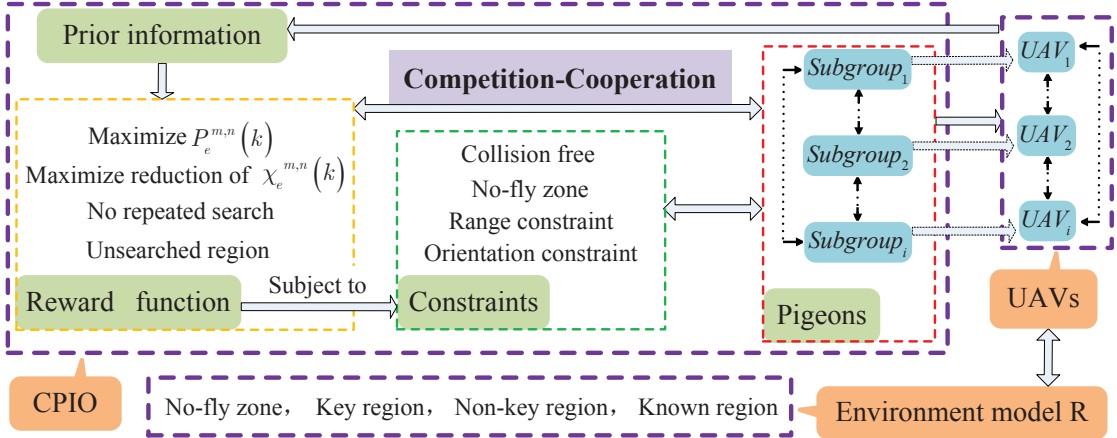

**Figure 4.** Schematic diagram of MUCS based on cooperation-competition mechanism.

Let $fitness(\cdot) = J_1$, thus Equation (15) can be transformed into:

$$L_{center}^{v-1} = \frac{\sum\limits_{a=1}^{C_2^{v-1}} L_k^{v-1} \cdot J_1(L_a^{v-1})}{C_2^{v-1} \cdot \sum\limits_{a=1}^{C_2^{v-1}} J_1(L_a^{v-1})}. \tag{17}$$

In the searching region, $L_a^u$ and $L_a^v$ are neighbor cells of $L_a^{u-1}$ and $L_a^{v-1}$, respectively. Under the constraint of Equation (6), the position (orientation) in the PIO algorithm can be expressed as:

$$L_a^u = \{(L_a^{u-1} - 1) \bmod 8, L_a^{u-1} \bmod 8, (L_a^{u-1} + 1) \bmod 8\}, \tag{18}$$

$$L_a^v = \{(L_a^{u-1} - 1) \bmod 8, L_a^{v-1} \bmod 8, (L_a^{v-1} + 1) \bmod 8\} = L_a^{u-1} + rand \cdot (L_{center}^{u-1} - L_a^{u-1}). \tag{19}$$

The MUCS process based on the CPIO algorithm can be expressed as Algorithm 1.

---

**Algorithm 1** MUCS based on CPIO algorithm

---

**Input:** Initializing environmental information and pose information of $UAV_i$.
**Output:** The searching trajectory of every UAV and its searching time steps $T_{i,1}(k)$.
1: **begin:**
2:     **for** $k = 1, ..., T_i$ **do**
3:         Predict N time steps in advance, and get the alternative paths.
4:         Delete the paths that does not satisfy Equation (8).
5:         Use the CPIO algorithm to find the path with the highest fitness value among the remaining paths.
6:         **while** $UAV_i$ receives the return order, $UAV_i$ returns to the airport.
7:         Record the searching trajectory and searching time steps $T_{i,1}(k)$ when $UAV_i$ returns.
8:         **end while**
9:     **end for**
10: **end**

---

## 4. ST Approach

Inspired by the knowledge of cooperative search and trajectory tracking [34,35], we propose the ST approach. The lowest $path_a$ without the OC of motion is mapped to a trajectory to be followed, and the cells it passes through are modeled as the key regions. Other blank cells are shaped as non-key regions, and obstacles are designed as no-fly zones. The input signal is the orientation sequence to maximize the search rewards of $UAV_i$. The process of tracking $path_a$ is the same as that of a UAV searching from the starting cell to the goal cell. Since the ST approach is only applied to a single UAV, there is no need to consider the cooperation-competition relationship between UAVs. ST approach based on basic PIO algorithm consists of four operators.

### Operator 1: Obtain $path_a$

We use A* algorithm [23] to find $path_a$, and A* algorithm can be simplified as the following steps:

Step 1: Specify the OC for the UAV.
Step 2: Design a cost function $f(m, n) = g(m, n) + h(m, n)$, where $g(m, n)$ denotes the movement cost: corresponds to the expenditure of moving the current position $(m, n)$ to other cell moved into the neighbor. $h(m, n)$ denotes the heuristic cost: corresponds to the expenditure of changing from current cell to goal cell. When $g(m, n) = 0$, A* algorithm degenerates to Dijkstra algorithm. When $h(m, n) = 0$, A * algorithm degenerates to greedy best first search algorithm.
Step 3: Estimate total expenditure $h(m, n)$ and change to the cell with the least cost.
Step 4: Repeat Step 3 until the goal cell is reached.
Step 5: When reaching the goal, choose the final path with least cost.

### Operator 2: Mark key cells

Referring to the concept of the TEP in the region searching, the blank cells are set as the non-key regions, which implies $P_{e,i}^{m,n}(k) = 0$. The cells that the path of $UAV_i$ $path_{a,i}$ passes through are marked as the key regions, and their $P_{e,i}^{m,n}(k)$ are denoted as follows:

$$P_{e,i}^{m,n}(k) = \begin{cases} \dfrac{1}{5}, & 1 < s_i < N_{m,i}, N_{f,i} = 0, \\ 1, & s_i = N_{m,i}, N_{f,i} = 0, \\ 0, & N_{f,i} \geq 1, \\ 0, & for\ other\ UAV_j(i \neq j), \end{cases} \tag{20}$$

where $s_i = 1, 2, ..., N_{m,i}$ is the serial number of the mark points from the starting cell to the goal cell in $path_{a,i}$. $s_i = N_{m,i}$ stands for goal point. When the key cell is searched for $N_{f,i} = 0$ time, the existence probability of the goal cell is set to be 1, and the probability of the other key cell is $\dfrac{1}{5}$. When $N_{f,i} \geq 0$, the probability of the corresponding key cell becomes 0.

**Remark 2.** *As shown in the fourth term of Equation (20), $UAV_i$ will only track $path_{a,i}$. The key cells marked for other UAVs are non-key cells for $UAV_i$. This ensures that the tracking process of the UAVs does not affect each other.*

### Operator 3: Design fitness function

The reward function of the ST approach relates only to the TEP for every cell in Operation 2:

$$J_1 = \sum_{q=1}^{N_p} P_e^{m,n}(k+q). \tag{21}$$

The original cell of $path_a$ is the starting position of the ST approach. And its initial flight orientation is pointed from the starting position to the second cell of $path_a$.

### Operator 4: Track $path_a$

The process of tracking is similar to that of Algorithm 1 and can be described as the following steps:

Step 1: Initialize the information of the region to be searched and the pose of the UAVs.

Step 2: Execute $N_p$SAP, and then get the predicted state sequence $\mathcal{E}_i(k)$.

Step 3: Let $fitness(\cdot) = J_2$, the path $p_i^l(k)$ corresponding to the sequence with the greatest fitness in $\mathcal{E}_i(k)$ solved by basic PIO.

Step 4: $UAV_i$ flies forward for a time step.

Step 5: Repeat Step 2 to Step 4 until the goal cell is searched.

Step 6: When reaching the goal, choose the final path with least cost and OC, as well as the total time steps $T_{i,2}(k)$ from the starting point to the goal point.

In light of the knowledge of graph theory, the key cells included by $path_a$ is the only directed connection sequence from the starting position to the goal cell, which is equivalent to a directed path. Through the four operations described above, we can obtain a lowest path that satisfies 3OC. ST approach as opposed to path following or trajectory tracking. The errors of the latter two are the absolute errors between the actual tracking trajectory and the ideal trajectory. ST approach chooses the path with the highest reward in all alternative paths, and the resulting tracking errors are the relative errors. Minimizing tracking errors can be converted to maximizing search rewards and it can be represented as:

$$\max J_2 = \max fitness(\mathcal{E}_i(k)) = fitness(p_i^*(k)) = \min \parallel \mathcal{L}_r(k) - \mathcal{L}_c(k) \parallel_2$$
$$= \min(\sum_{q=1}^{N_p} \sqrt{(x_r(k+q) - (x_c(k+q))^2 + (y_r(k+q) - (y_c(k+q))^2}, \tag{22}$$

where $p_i^*(k)$ denotes the path with the greatest fitness in $\mathcal{E}_i(k)$ at time step $k$. $\mathcal{L}_r(k) = (x_r(k), y_r(k))$ is the coordinate of the key cell to be tracked. $\mathcal{L}_c(k) = (x_c(k), y_c(k))$ is the coordinate of the cells actually tracked.

If $\mathcal{L}_r(k) \in \mathcal{E}_i(k)$, the tracking process with the highest search rewards must be error-free tracking. Algorithm 2 demonstrates the implementation procedure of the ST approach.

**Remark 3.** *(1) ST approach is not only suitable for A\* algorithm. It is effective for other algorithms. (2) In addition to the 3OC mentioned in this section, the path can also be tracked under 4OC and 8OC. ST approach is suitable for different types of robots or different environments. (3) This approach can also track the 3D path or the path under the dynamic environment.*

---

**Algorithm 2** ST approach

---

**Input:** The lowest $path_{a,i}$ without the OC.
**Output:** The lowest $path_{b,i}$ with the OC, the tracking time steps $T_{i,2}(k)$.
1: **begin:**
2:    Use the A* algorithm to get $path_{a,i}$.
2:    Mark the key cells contained in $path_{a,i}$.
3:    **for** $m = 1, ..., L_x$
4:      **for** $n = 1, ..., L_y$ **do**
5:        Assign different $P_{e,i}^{m,n}(k)$ to the cells of whole map in term of Equation (23).
6:      **end for**
7:    **end for**
8:    Let $fitness(\cdot) = J_2$.
9:    **for** $k = 1, ..., T_i'(k)$(unknow)
10:      Use basic PIO algorithm to track $path_{a,i}$ under OC.
11:      **while** the goal cell (airport) is reached.
12:        Record the return $path_{b,i}$ and $T_{i,2}(k)$.
13:      **end while**
14:    **end for**
15: **end**

---

## 5. DTSCS Scheme for MUCS

According to the analysis and requirements of Section 2.3, we design the DTSCS scheme for closed search. In the first stage, the CPIO algorithm proposed in the Section 3 is used as the search algorithm for MUCS. In the return stage, the ST approach proposed in the former section is used to obtain the shortest path with OC to return to the airport. The two stages are coupled in time. The pose of UAV at the end of the search stage is the starting pose for the return stage, which in turn determine the time steps needed for UAV to return to the starting airport.

In the search stage, $UAV_i$ needs to calculate its remaining distance and the time steps needed to return to the airport in real time. Assume the RC of $UAV_i$ is $T_i$. $T_{i,1}(k)$ represents the total time steps of the search stage. $T_{i,2}(k)$ is the total time steps of the return stage. Assume that $UAV_i$ has searched for $k$ time steps and $UAV_i$ can still safely return to the airport. Before executing the next-step search, $UAV_i$ need first to calculate the position to be reached at the next step and how long it will take from the current cell returning back to the airport. If at the next step, the calculated fuel can still guarantee $UAV_i$ safely returning back to the airport, then the $UAV_i$ search forward a time step. Otherwise, the $UAV_i$ stop searching and return directly to the airport. Therefore, the time for $UAV_i$ returning to the airport should satisfy the following relationships:

$$T_{i,1}(k) + T_{i,1}(k) \leq T_i, \tag{23}$$

$$T_{i,1}(k+1) + T_{i,1}(k+1) > T_i, \tag{24}$$

where Equation (23) means that the sum of $T_{i,1}(k)$ and $T_{i,2}(k)$ at any time step cannot exceed the RC. Equation (24) specifies the condition for the return of the $UAV_i$. The DTSCS scheme can not only ensure that UAV gets the maximum search rewards, but also maximize the range. Besides, it is not limited to the fixed range $T_i$. It can be changed during the task. We can specify the searching time for the first stage, or we can issue a return order at any time during the search. All of these scenarios are possible cases in the application of MUCS.

**Remark 4.** *Under return conditions, UAV may have surplus fuel after returning to the airport, which is due to the OC of itself.*

**Definition 5. (Range utilization)**. *The proportion of time-of-flight of UAV in $T_i$ is defined as:*

$$\eta_i = \frac{T_{i,1}(k) + T_{i,2}(k)}{T_i} \times 100\%. \tag{25}$$

Set the RC for the $UAV_i$ to be 100 steps. Taking an example shown in Figure 5, the remaining range at this time step is assumed to be six steps. If $UAV_i$ continues to search, it will take at least nine steps, as shown in black arrow, for $UAV_i$ to return to the airport at the next step. According to Equations (23) and (24), $UAV_i$ should return to the airport at this time step, as shown in red arrow. In this case, $\eta_i = 95\%$. Algorithm 3 describes the detailed flows of the DTSCS scheme.

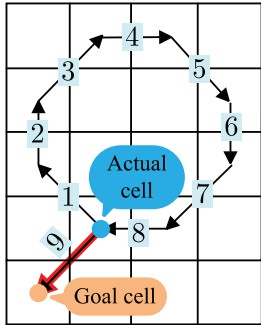

**Figure 5.** Cause for the remaining range.

---

**Algorithm 3** DTSCS scheme

---

**Input:** Initializing environmental information, pose information of $UAV_i$, and $T_i$.
**Output:** $T_{i,1}(k)$, $T_{i,2}(k)$, closed trajectory and $\eta_i$.
1: **begin:**
2:     **for** $k = 1, ..., T_i$ **do**
3:         **if** $T_{i,1}(k) + T_{i,1}(k) \leq T_i$ and $T_{i,1}(k+1) + T_{i,1}(k+1) \leq T_i$
4:           **do** the search stage (Algorithm 1).
5:         **else if** $T_{i,1}(k) + T_{i,1}(k) \leq T_i$ and $T_{i,1}(k+1) + T_{i,1}(k+1) > T_i$
6:           **do** the return stage (Algorithm 2).
7:         **while** the airport is reached.
8:         **end while**
9:     **end for**
10: **end**

---

## 6. Numerical Simulation and Analysis

In this section, three simulations are performed to verify the performance of the proposed CPIO and the effectiveness of the ST approach and the DTSCS scheme. The simulation programs are coded in Matlab R2016a and implemented on Intel Core I7-2600 3.40 GHz personal computer with 4 GB random access memory.

### 6.1. MUCS Based on CPIO Algorithm

The region $R$ consists of $50 \times 50$ cells, the center of every cell is the allowed waypoint. And the setting of the coordinate system of $R$ is consistent with Figure 9a. The starting positions, i.e, the locations of starting airports, and orientations of the four UAVs are as follows.

$UAV_1$: [(1, 20), 2] where (1, 20) is the position, and 2 is the orientation, $UAV_2$: [(20, 50), 4], $UAV_3$: [(27, 1), 0], $UAV_4$: [(50, 26), 6]. No-fly zones: $5 \leq x \leq 20$, $30 \leq y \leq 40$ and $30 \leq x \leq 40$, $5 \leq y \leq 15$. Known region: $30 \leq x \leq 45$, $20 \leq y \leq 28$, where $P_e^{m,n}(k) = 0$, $\chi_e^{m,n}(k) = 0$. Generating 5 randomly distributed targets shown in pink stars in each key region: $5 \leq x \leq 24$, $10 \leq y \leq 25$ and $25 \leq x \leq 45$, $30 \leq y \leq 35$, where $P_e^{m,n}(k) = 0.9$, $\chi_e^{m,n}(k) = 0.9$. Other cells are non-key regions, where $P_e^{m,n}(k) = 0.5$, $\chi_e^{m,n}(k) = 0.5$. The other simulation parameters are listed in Table 1.

**Table 1.** Parameters in MUCS.

| Parameters | Description | Value |
|---|---|---|
| $P_c^{m,n}(k)$ | Detection probability of the UAV sensor to the goal. | 0.9 |
| $P_f^{m,n}(k)$ | False alarm rate. | 0.1 |
| $\omega_1, \omega_2, \omega_3, \omega_4$ | Weight factor of fitness function $J_1$. | 0.3, 0.3, 0.3, 0.1 |
| $M$ | Number of UAVs (subgroups). | 4 |
| $C_1$ | Initial number of pigeons in every subgroup. | 40 |
| $R_p$ | Coefficient of the map and compass operator. | 0.5 |
| $N_{c1}, N_{c2}$ | Number of iterations of Operation 1 and Operation 2 in the CPIO algorithm. | 25, 20 |

**Scenario 1:** Search for the best $N_p$

The advantages of $N_p$SAP are discussed in Section 3.1. If $N_p$ is relatively large, the benefits of $N_p$SAP will be limited or even negligible. In contrast, the number of alternative paths will increase exponentially. Figure 6 shows the simulation results and error analysis with different values of $N_p$.

In Figure 6a, the running time $T_a$ of the personal computer increases with the increase of $N_p$. When $N_p = 7$ and the time steps of the search is 100, the computation time is $T_a = 121.7$ s. Combined with the actual search process, it do not meet the requirements of the real-time search task. Figure 6b shows that if $N_p = 1$ or 2, the average fitness value, average rewards for the search process, decreases gradually in the later stages of the search, this is because one or two time steps in advance cannot effectively avoid obstacles or continue to search within the key region. When $N_p = 4$, UAV has enough time to make obstacle avoidance and continue to search in the key regions. However, the increase of $N_p$ will not significantly improve the rewards. In Figure 6c, when $N_p \leq 4$, the number of targets found out is less instead. When $N_p = 5$ to 7, the number of targets found out are the same. Based on the above analysis, we choose $N_p = 5$ as the best prediction steps. Figure 7 shows the search trajectories of four UAVs when $N_p = 5$.

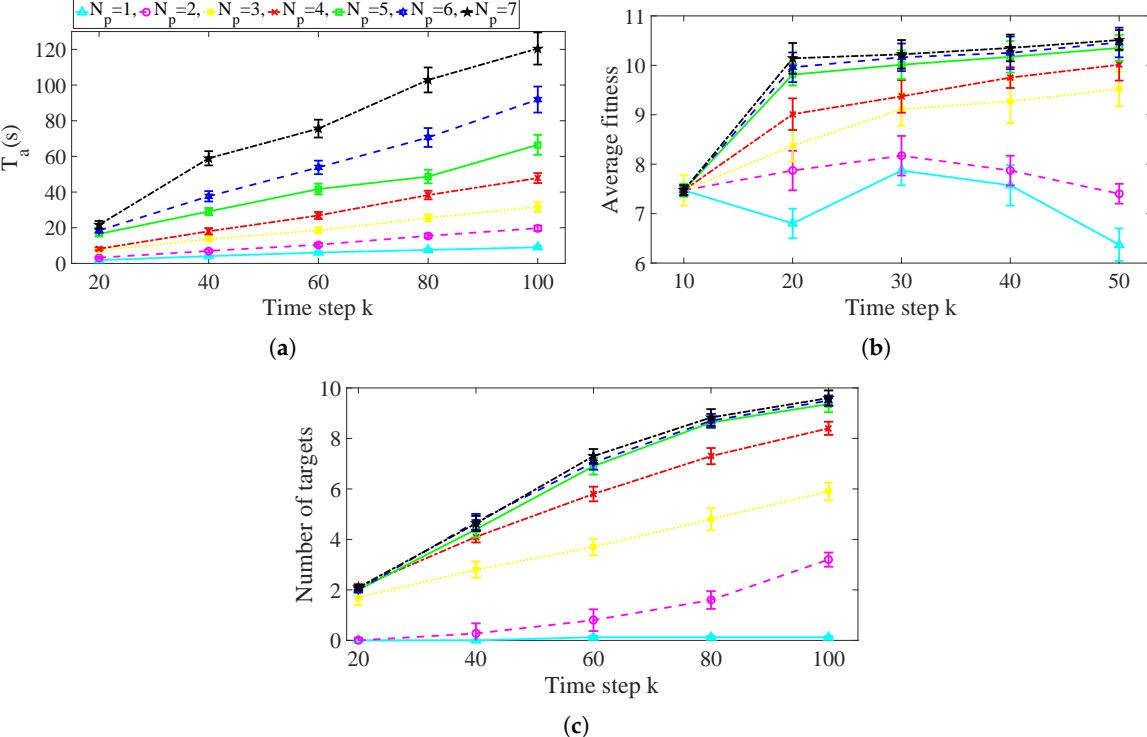

**Figure 6.** Error analysis for different values of $N_p$: (**a**) Running time; (**b**) Average fitness value; (**c**) Number of targets found.

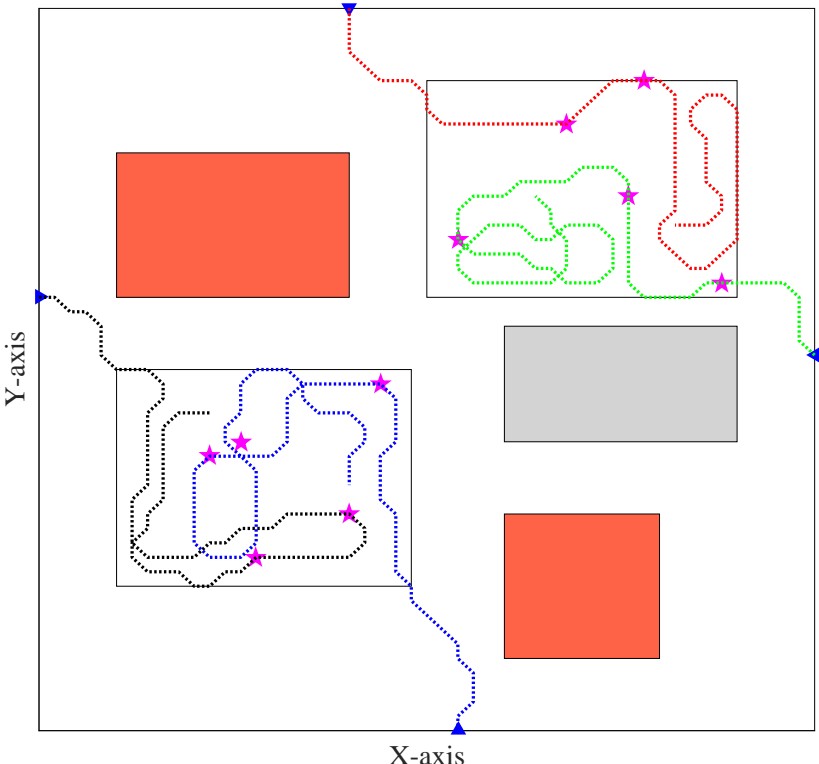

**Figure 7.** Search trajectory with $N_p = 5$.

**Scenario 2:** Searching performance of CPIO

To demonstrate the effectiveness of CPIO, comparative experiments are conducted with identical initial conditions. The searching performance of CPIO is compared to the basic PIO, particle swarm optimization (PSO), and GA. In Figure 8a, The convergence speed of the CPIO algorithm is faster than the basic PIO, PSO and GA. And the standard deviations of 100 experiments are also the smallest. Figure 8b shows that when the searching time step is 100, the average number of targets found out by CPIO is 9.6, which is more than the other three algorithms. Therefore, the CPIO algorithm based on cooperation-competition mechanism is superior to the compared algorithms in the cooperative search task.

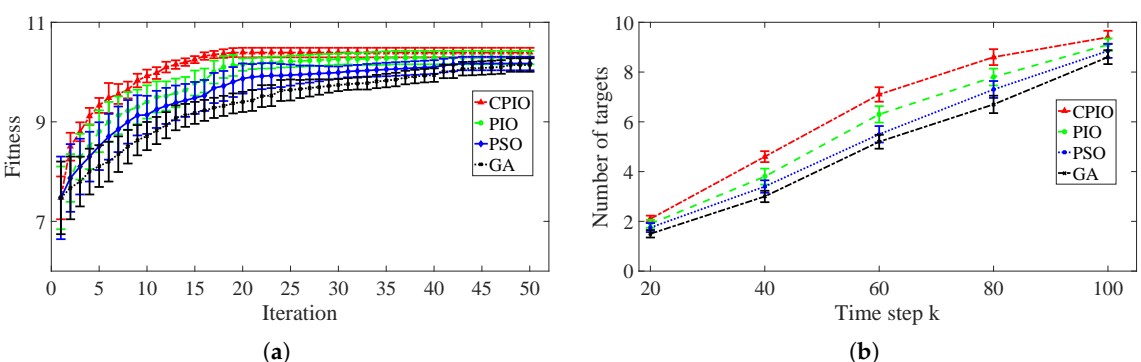

**Figure 8.** Search performance of CPIO, PIO, PSO, and GA: (**a**) Convergence speed; (**b**) Average number of targets found out.

### 6.2. Tracking Performance of the ST Approach

**Definition 6. (Tracking efficiency $\phi$).** *The proportion of the key cells tracked in the total cells tracked by ST approach, which is the indicator of tracking performance.*

The cells that A* algorithm passes through are modeled as the key regions, and Figure 9a shows the marked results. Using basic PIO algorithm to track these key cells, the tracking results under 3OC, 4OC, and 8OC are shown in Figure 9b. In some slit areas, $\phi$ will be reduced because the key cells cannot be tracked completely under the 3OC. The A* algorithm and Dijistra algorithm can not satisfy the 3OC in Area 1 and Area 2. To maximize the rewards, there will be some adaptive path selections, which via a few steps ahead of the turn or lag a few steps back to track the key cells. The $\phi$ for different OCs are given in Figure 10. Its abscissa represents the unit length after subdividing Figure 9a. The search efficiency of 8OC is higher than that of 3OC and 4OC. As the grid map is subdivided, The $\phi$ of 3OC and 4OC increases.

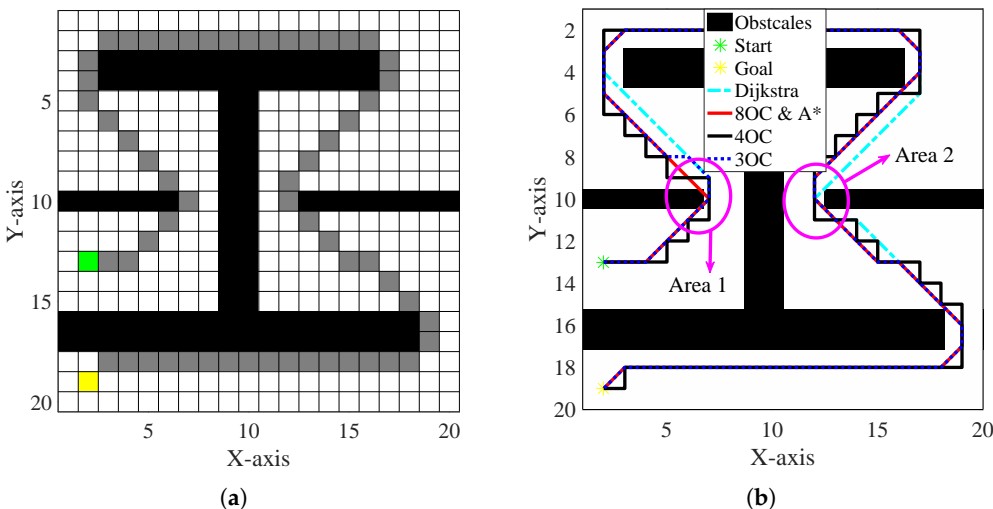

(a)　　　　　　　　　　　　　　　　　　　(b)

**Figure 9.** Effect drawing of the ST approach, A* algorithm and Dijkstra algorithm: (**a**) Marked key cells; (**b**) Result of tracking.

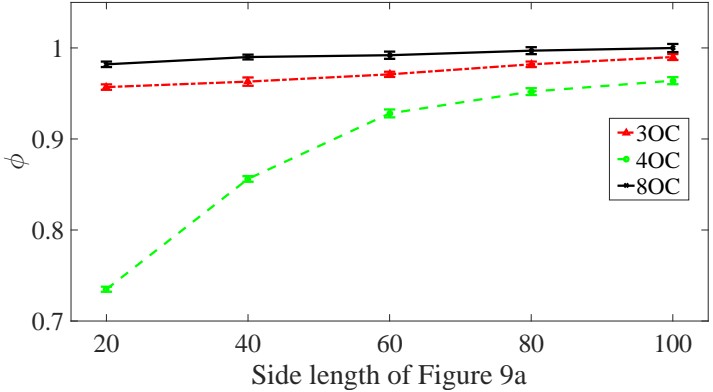

**Figure 10.** Search efficiency under different OCs.

### 6.3. Closed Trajectory

The numerical simulation is carried out according to the DTSCS scheme designed by Section 5. Let $T_{i,1}(k) = 100$. The closed trajectory is shown in Figure 11, where the dotted line is the search path, and the solid line is the return path. Figure 12 shows the relationship between range utilization and time step for every UAV. The average range utilization of UAVs is 97% under the 3OC. If switching to the 8OC, the range utilization of every UAV will increases.

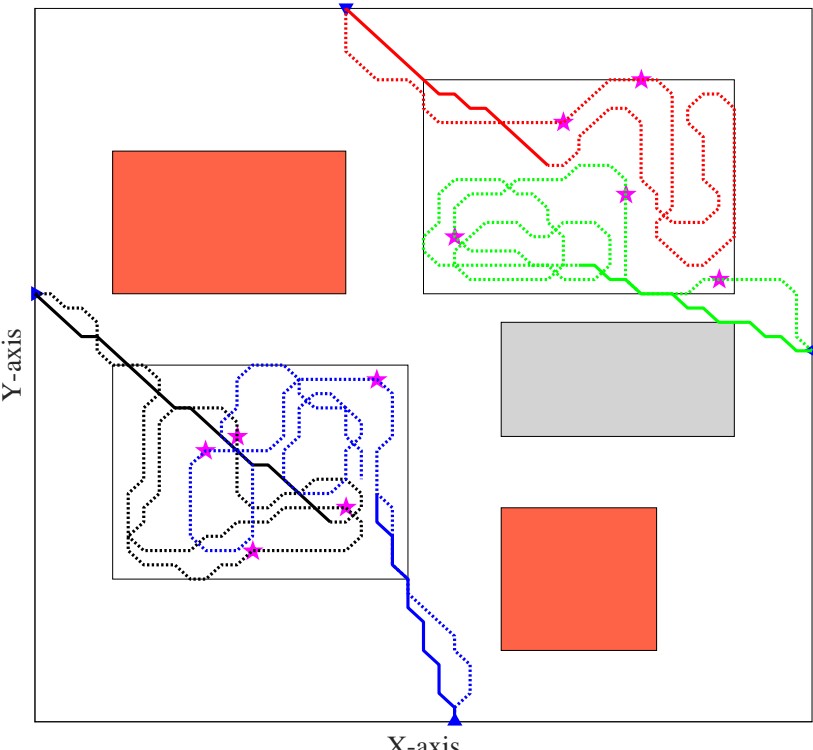

**Figure 11.** Closed trajectory of UAVs.

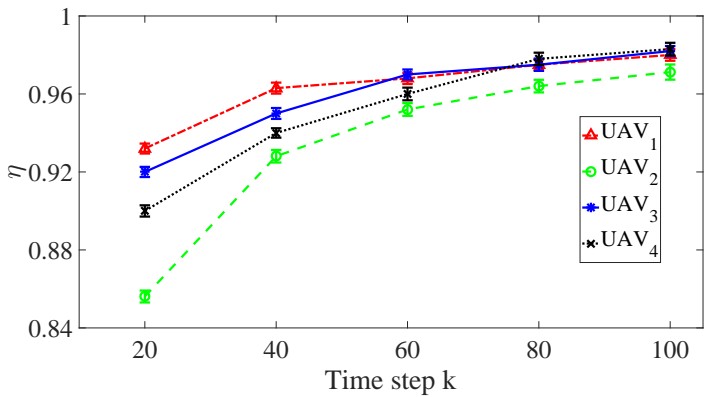

**Figure 12.** Range utilization of UAVs.

## 7. Conclusions

According to the RC of UAV in the search process, we design a dynamic two-stage scheme to implement the closed search. Every UAV needs to take into account the time steps needed to return to the airport during searching process. When the searching time and the returning time meet the Equations (23) and (24), UAVs can return back to the starting airports. To improve the target search efficiency, the CPIO with cooperation-competition mechanism is proposed and applied to the search stage problem. The simulation results show that $N_p = 5$ is the best prediction time steps. The CPIO algorithm outperforms the compared algorithms regarding the number of targets found out and the convergence speed. In the return stage, the ST approach is presented to ensure UAVs safely return to the their starting airport. Closed target searching simulations demonstrate the effectiveness and efficiency of the proposed algorithm. In the future, we will explore the cooperative search strategy and path planning in dynamic 3D environments.

**Author Contributions:** Conceptualization, D.L. and J.S.; methodology, D.L.; software, J.S.; validation, D.L., J.S., Y.X., Y.Y. and H.D.; formal analysis, J.S.; investigation, Y.X.; resources, D.L.; data curation, D.L., Y.X., Y.Y. and H.D.; writing–original draft preparation, J.S.; writing–review and editing, D.L., Y.X., Y.Y. and H.D.

**Funding:** This work is supported by the National Natural Science Foundation of China under Grant (No. 61673327) and the Natural Science Foundation of Fujian Province of China under Grant (No. 2016J06011).

**Conflicts of Interest:** The authors declare no conflict of interest.

## Abbreviations

The following abbreviations are used in this manuscript:

| | |
|---|---|
| UAV | unmanned aerial vehicle |
| MUCS | multi-UAV cooperative search |
| DTSCS | dynamic two-stage closed search |
| ST | search tracking |
| RC | range constraint |
| OC | orientation constraint |
| $N_p$SAP | $N_p$-step-ahead prediction |
| TEP | target existence probability |
| EU | environmental uncertainty |
| PIO | pigeon-inspired optimisation |
| CPIO | coevolution pigeon-inspired optimization |
| 3D | three-dimensional |

## Appendix A

Here, $d_{ij}(k)$ is described as (A1)

$$d_{ij}(k) = \begin{bmatrix} d_{11}(k) & d_{12}(k) & ... & d_{1M}(k) \\ d_{21}(k) & d_{22}(k) & ... & d_{2M}(k) \\ ... & ... & ... & ... \\ d_{M1}(k) & d_{M2}(k) & ... & d_{MM}(k) \end{bmatrix} \tag{A1}$$

where $d_{ij}(k) = \sqrt{(x_i(k) - x_j(k))^2} + \sqrt{(y_i(k) - y_j(k))^2}$ denotes the Euclidean distance of $UAV_i$ and $UAV_j$. $M$ is the number of UAVs. $d_{ii}(k) = \infty$. $(x_i(k), y_i(k))$ and $(x_j(k), y_j(k))$ are the positions between $UAV_i$ and $UAV_j$ in time step $k$.

$d_{ih}(k)$ and $d_{ij}(k)$ are similar:

$$d_{ij}(k) = \begin{bmatrix} d_{11}(k) & d_{12}(k) & ... & d_{1H}(k) \\ d_{21}(k) & d_{22}(k) & ... & d_{2H}(k) \\ ... & ... & ... & ... \\ d_{M1}(k) & d_{M2}(k) & ... & d_{MH}(k) \end{bmatrix} \tag{A2}$$

where $d_{ih}(k) = (k)\sqrt{(x_i(k) - x_h(k))^2} + \sqrt{(y_i(k) - y_h(k))^2}$ is the Euclidean distance between $UAV_i$ and airport$_h$ ($h = 1, 2, ..., H$), $H$ is the number of airport.

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
