# Peer review of "Coevolution Pigeon-Inspired Optimization with Cooperation-Competition Mechanism for Multi-UAV Cooperative Region Search"

_applsci, doi:10.3390/app9050827_

Round 1
Reviewer 1 Report
The authors describe a method for multi-UAVs systems to explore areas of interest in an efficient manner.
Comments:
In general terms, the article will benefit from a review by a native speaker.
Authors claim in the contributions that they are the first considering "closed search". What do you mean in this context? I read several articles that consider the requirement of going back to the base station in the mission.
Sections 2.3.2 introduces a set of restrictions, but I cannot see how these are traduced to mathematical constraints in the cost function.
Section 3.2 introduces PIO. But why is the explanation in 3D if the problem of interest is in 2D?
Figure 6 is a bit crowd, and the lines share color have similar markers.
In figures 9, what is the particular improvement compared to any other path planning algorithm?
Minor comments:
Line 1 "scheme for the unmanned aerial vehicles (UAV) is designed"
In line 33 what do you mean with "no targeted research has been conducted"
Line 91 is very confusing
line 91-93 mention "P_e" and "X_e" but they are not introduced properly
Author Response
Dear Editor-in-chief, Associate Editor and Anonymous Reviewers,
Re: Manuscript reference ID: applsci-439755
We would like to express our heartfelt thanks to you and the reviewers for the highly
insightful comments which enable us to greatly improve the quality of our manuscript. we
have carefully revised our paper. In the following pages are our point-by-point responses
to each of the comments of the reviewers as well as your own comments.
Please find attached a revised version of our manuscript “Coevolution Pigeon-inspired
Optimization with Cooperation-competition Mechanism for Multi-UAV Cooperative Region
Search”, which we would like to resubmit for publication as Regular paper in Applied
Sciences.
In accordance with two reviewers’ suggestion, we fix these grammatical typos to the
best of us and add the necessary discussion. We hope that the revisions in the manuscript
and our accompanying responses will be sucient to make our manuscript suitable for
publication in Applied Sciences.
We shall look forward to hearing from you at your earliest convenience.
Yours sincerely,
Associate Prof. Luo Delin
Address: School of Aerospace Engineering, Xiamen University
E-mail: luodelin1204@xmu.edu.cn

Reviewer 2 Report
This paper presents a closed search algorithm for UAVs composed of two stages: search stage and return stage. For the search stage, a coevolution pigeon-inspired optimization is proposed, and for the return stage a search tracking algorithm is presented. The paper is well structured and has a lot of figures to illustrate the algorithms and the results presented but, nevertheless, I think is necessary to modify some important points of the paper before its publication.
In the abstract you present as a result the “full use of the fuel”. This affirmation must be supported by the simulations and must be included in the section 6. Furthermore, this is not consistent with remark 3.
Also, in the abstract, the affirmation made in the second result must be completed.
In the introduction, in line 33, the writing motivation of the paper does not seem to be the study of search strategies under RC and OC rather the proposal of a new strategy.
In the introduction, in line 33 too, I do not understand the sentence “To our knowledge, however, no targeted research has been conducted.”
In some lines (49, 51,…) you use the term robot instead of UAV. Unify terms.
The affirmation you make in line 55 it is not clear, specially reading section 2.3.1. There are a lot of papers that take into account the return to the base station of the UAVs. This affirmation must be cleared.
For CPIO, every subgroup represents a UAV. This could be a very strong simplification. In your paper, in line 147, you described the algorithm as inspired by cooperation-competition relationship between subgroups.
In line 99, the term Nf(m,n) must be described in more detail.
In section 4, you use the A* algorithm for ST. Why do not use another algorithm like D* or similar? A* does not take into account changes in the search environment. How can you avoid another UAVs that are coming back to the base station too?
In lines 363 and 364 you use the acronyms SO an GA. Please, use the complete names before use acronyms.
Author Response

(The authors gave the same response as above.)

Round 2
Reviewer 1 Report
The authors did a great work tacking the comments.
The article has been greatly improved with authors changes and I consider it to be accepted in its current form.
Kind regards,
Reviewer 2 Report
In this new version, the paper has been improved notably and it is suitable for its publication.